# Manipulation of Oxidative Stress Responses by Non-Thermal Plasma to Treat Herpes Simplex Virus Type 1 Infection and Disease

**DOI:** 10.3390/ijms24054673

**Published:** 2023-02-28

**Authors:** Julia Sutter, Peter J. Bruggeman, Brian Wigdahl, Fred C. Krebs, Vandana Miller

**Affiliations:** 1Center for Molecular Virology and Gene Therapy, Institute for Molecular Medicine and Infectious Disease, Department of Microbiology and Immunology, Drexel University College of Medicine, Philadelphia, PA 19129, USA; 2Department of Mechanical Engineering, University of Minnesota, Minneapolis, MN 55455, USA

**Keywords:** oxidative stress, immunomodulation, reactive oxygen and nitrogen species, redox homeostasis, antioxidant, innate immunity, adaptive immunity, antiviral therapy, plasma

## Abstract

Herpes simplex virus type 1 (HSV-1) is a contagious pathogen with a large global footprint, due to its ability to cause lifelong infection in patients. Current antiviral therapies are effective in limiting viral replication in the epithelial cells to alleviate clinical symptoms, but ineffective in eliminating latent viral reservoirs in neurons. Much of HSV-1 pathogenesis is dependent on its ability to manipulate oxidative stress responses to craft a cellular environment that favors HSV-1 replication. However, to maintain redox homeostasis and to promote antiviral immune responses, the infected cell can upregulate reactive oxygen and nitrogen species (RONS) while having a tight control on antioxidant concentrations to prevent cellular damage. Non-thermal plasma (NTP), which we propose as a potential therapy alternative directed against HSV-1 infection, is a means to deliver RONS that affect redox homeostasis in the infected cell. This review emphasizes how NTP can be an effective therapy for HSV-1 infections through the direct antiviral activity of RONS and via immunomodulatory changes in the infected cells that will stimulate anti-HSV-1 adaptive immune responses. Overall, NTP application can control HSV-1 replication and address the challenges of latency by decreasing the size of the viral reservoir in the nervous system.

## 1. Introduction to HSV-1 Infection

As a prevalent pathogen and global health concern, herpes simplex virus type 1 (HSV-1) is among one of the most studied human herpesviruses. HSV-1 has long served as a useful research tool in providing information that has enhanced the understanding of viruses. It also has a large global footprint in which more than 70% of the world’s population under the age of 50 harbors an HSV-1 infection [1]. Most individuals who are infected with HSV-1 are asymptomatic because the viral genome remains transcriptionally dormant within the peripheral nervous system during latent infection, allowing lifelong persistence of the virus. Replication of this latent virus can be reactivated by external and internal stress stimuli, resulting in productive infection in neurons and re-infection of epithelial cells in the mucosal epithelium innervated by the infected neuron. Reactivation results in clinical signs of infection in the form of cold sores around the mouth, eyes, and genitalia. Although these lesions can be discomforting to the patient, most cases of HSV-1 infection are mild [2,3]. In rare cases, HSV-1 can cause serious disease by spreading to the brain to cause encephalitis [4] and to the eye to cause keratitis [5]. There is also increasing evidence that chronic HSV-1 infection can contribute to the development of neurodegenerative diseases such as Alzheimer’s Disease later in life [6,7,8] (Figure 1).

For patients who experience severe clinical symptoms of HSV-1 infection, treatment options, while available, are limited and sometimes ineffective. Antiviral therapies are typically administered upon the appearance of clinical symptoms after the initial acquisition of the virus or when the virus is reactivated from latency. The first line of defense against these symptoms is nucleoside analogs, the most common being acyclovir. As a prodrug, acyclovir becomes activated through phosphorylation by virus-encoded thymidine kinases (TK) to the monophosphate form and further phosphorylation by other thymidylate kinases, converting it to acyclovir trisphosphate [9,10,11]. This form of acyclovir then competes with guanine nucleotides and binds the growing DNA strand during viral DNA replication in the nucleus, thus inhibiting the synthesis of the viral DNA. This leads to the selective inhibition of the viral DNA polymerase [11]. Penciclovir is another nucleoside analog prodrug prescribed to patients with a similar mechanism of action as acyclovir, but with an even higher affinity for the viral DNA [10,11].

Due to their mechanism of action, nucleoside analogs are effective during acute infection but do not affect the latent virus residing in neurons, allowing HSV-1 to persist in its host. This persistence can lead to the development of viral mutants upon reactivation, which give rise to drug-resistant strains [9,11,12]. Additionally, by failing to eliminate the virus, these antiviral therapies do not prevent future recurrent acute infection and transmission of HSV-1 that occurs through direct contact with the resulting lesions [1,3]. These limitations highlight the need for new therapeutic approaches that effectively address viral latency and reactivation by targeting other available viral and cellular components involved in HSV-1 pathogenesis. These include structural components of the virion structure and cellular processes manipulated by HSV-1 to facilitate productive infection of the cell.

Reduction-oxidation (redox) homeostasis, which involves the maintenance of intracellular reactive oxygen and nitrogen species (RONS), is one of these processes that is significantly disrupted during HSV-1 infection in both epithelial cells (host cells for lytic infection) and neurons (latently infected cells). Disruption of redox homeostasis by HSV-1 in both cell types allows the increase in RONS generation, leading to greater RONS concentrations in spite of secondary reactions and enzymatic pathways. This increase in RONS, as a result, contributes to their activity in modulating cellular signaling pathways, the downregulation of cellular antiviral responses, and the promotion of an environment favorable for HSV-1 infection [13]. This review discusses how the development and progression of HSV-1 to chronic infection is closely tied to the alteration of the redox state in the host cell and subsequent reductions in the cell’s antiviral defense against HSV-1. Since maintaining redox homeostasis is critical in cells, we discuss how alterations in cellular redox by the virus may be harnessed as a therapeutic target for HSV-1 infections.

## 2. HSV-1 Disrupts Cellular Redox Homeostasis during Infection

RONS are potent intracellular signaling molecules with abilities to stimulate various cell signaling pathways, including those that regulate cell proliferation and apoptosis [14,15,16,17,18,19]. RONS also modulate immune responses against invading pathogens through the manipulation of cell signaling, or by influencing the production of proinflammatory mediators to promote an antiviral environment in and around the cell. Cellular RONS often participate in antimicrobial killing of invading pathogens during phagocytosis [20]. These functions of RONS are designed to maintain the health and integrity of the cell. HSV-1 is an obligate intracellular pathogen that relies heavily on host cellular machinery for its replication. To create an intracellular environment that favors productive replication (or sustained latent infection), HSV-1 affects multiple cellular processes, including redox homeostasis.

### HSV-1 Can Hijack Cellular RONS Generation to Cause Oxidative Stress

To maintain redox homeostasis, RONS are constantly produced and destroyed in a healthy cell. The functions of RONS are attributed to their abilities to react with macromolecular structures. RONS are reactive species derived from oxygen and/or nitrogen-containing compounds, some containing an unpaired electron in the outer shell [21,22]. Therefore, these species can modify macromolecules leading to the modulation of signaling pathways. These actions are characteristic of primary RONS such as nitric oxide, hydrogen peroxide, and superoxide [15]. In the cell, primary RONS are reactive species that are typically generated during normal metabolic processes that occur in the mitochondria [23,24,25,26,27,28,29,30], endoplasmic reticulum (ER) [31,32], and peroxisome [33,34]. Alternatively, these RONS can be generated via specialized enzymes present in the cell such as nicotinamide adenine dinucleotide phosphate (NADPH) oxidases (NOX) [35,36,37] or nitric oxide synthases (NOS) [38,39,40]. These primary RONS may alter signaling cascades by reversible modifications on target macromolecular structures. Overall, primary RONS exhibit a weak damaging potential. Due to their metastable nature, primary RONS can participate in reactions with one another to form secondary RONS (e.g., peroxynitrite and hypochlorous acid). Unlike primary RONS, secondary RONS are not tightly regulated by the cell and can inflict damage when they accumulate [41,42]. This accumulation and subsequent damage of the cell by RONS is referred to as oxidative stress.

Oxidative stress is often induced by viruses to overwhelm the infected cell and create an environment that favors its replication. For example, respiratory syncytial virus (RSV) upregulates RONS in airways, while simultaneously diminishing the host’s antioxidant system [21,43]. Similarly, upregulation of serum markers of lipid peroxidation in human immunodeficiency virus type 1 (HIV-1)-infected patients suggests increased oxidative stress and a link between RONS dysregulation and viral pathogenesis [44,45]. Oxidative stress in infected cells can damage cellular macromolecules and impair immune responses against the virus. However, the viral genome is also at risk for modification and mutation, leading to the formation of viral variants that may have enhanced virulence [46].

## 3. HSV-1 Lytic Infection Manipulates Oxidative Stress in the Cell

Like RSV and HIV-1, HSV-1 induces intracellular oxidative stress conditions during infection [47,48]. HSV-1 infection results in two distinct outcomes: lytic infection and latent infection. Lytic infection is characterized by active viral gene expression and results in the assembly of progeny virions that are released into the extracellular environment to infect nearby uninfected cells. During a lytic infection, the high level of replication results in the destruction of the host cell. This type of HSV-1 infection occurs in mucosal epithelial cells, usually at the site of infection or reactivation. During a period of acute infection characterized by lytic replication, patients present with cold sores and are responsive to antiviral drugs administered to alleviate these clinical symptoms [2,3]. Lytic infection depends on multiple viral and cellular components to establish and sustain infection in epithelial cells, impacting the entire cell and utilizing cellular processes to create an environment suitable for productive HSV-1 replication. One way this occurs is through the manipulation of oxidative stress responses by the virus, generating RONS that can interfere with cellular signaling and responses.

### 3.1. HSV-1 Alters Cellular Redox Homeostasis Early in Infection

HSV-1 is an enveloped virus, composed of an encapsulated genome surrounded by a host cell-derived lipid bilayer decorated with glycoproteins [49,50]. In the initial steps of a lytic infection, these envelope glycoproteins facilitate attachment and subsequent entry into target epithelial cells through interactions with cellular heparan sulfate (HS) [5,51,52,53], a cell surface proteoglycan involved in signaling, host defense, and metabolic processes [54]. Upon glycoprotein-receptor binding, cytoskeletal rearrangements are triggered to allow membrane fusion between the viral envelope and host cell membrane, facilitating the release of the protein capsid and the enclosed viral genome into the cytoplasm [5,51,52,53,55].

In establishing a redox environment that favors HSV-1 replication, two envelope proteins have secondary roles in recalibrating oxidative stress responses in the cell. Glycoprotein B (gB), which mediates attachment to HS, also induces ER stress by interacting with the ER lumen domain protein kinase R-like ER kinase (PERK), a signaling transducer involved in the ER stress response to misfolded proteins during viral infections. To prevent ER-mediated oxidative stress and subsequent apoptosis, gB interacts with PERK to maintain homeostasis in the ER [56]. HSV-1 glycoprotein J (gJ), which does this during binding and entry, is also implicated in viral-induced oxidative stress by promoting RONS accumulation through the adenosine triphosphate (ATP) synthase molecule [57,58].

Following fusion of the viral and cellular membranes, compartmentalization of the encapsulated viral genome into host endosomes allows trafficking of the viral contents through the cytoplasm toward the nucleus in a pH-dependent process. The lower pH of the endosome facilitates the release of the capsid into the cytosol [59]. HSV-1 tegument proteins, which are packaged in the virion, are simultaneously released into the cytoplasm where they can regulate cellular processes in favor of HSV-1 replication and subsequent pathogenesis [55,60,61,62]. Like the envelope proteins, HSV-1 tegument proteins influence the redox state of the cell to prevent clearance of the virus. Virus protein 16 (VP16), a tegument protein released into the cytoplasm following viral entry, manipulates HSV-1 by downregulating nuclear factor-κB (NF-κB) immune signaling, which is known to be responsive to RONS [60,63]. HSV-1 can also manipulate oxidative stress responses in the cell through infected cell protein 0 (ICP0), which is expressed as an immediate-early (IE) protein but is also packaged in the virus. By acting as an ubiquitin ligase, ICP0 can selectively target host cell signaling proteins to promote their degradation and loss of function [64]. The actions of these tegument proteins are critical in downregulating the antiviral response toward HSV-1, creating an intracellular environment suitable for productive replication. HSV-1 continues to hijack cellular machinery, such as the endocytic pathway, to allow the trafficking of the viral genome into the nucleus through the nuclear pore complex (NPC). Additionally, tegument proteins facilitate the shedding of the viral protein capsid and release of the genome into the nucleus [55,60,62,65,66,67,68].

Upon entry into the nucleus, the HSV-1 genome is still at risk of detection and subsequent clearance by the cell. Therefore, HSV-1 continues to employ tactics to dodge these attempts at cellular immune detection by manipulating redox homeostasis. By upregulating the production of RONS in the cell, HSV-1 can modify key cell signaling proteins to dampen and negate innate detection mechanisms. For example, HSV-1 interferes with the cyclic GMP-AMP synthase/stimulator of interferon genes (cGas/STING), a nuclear DNA sensing pathway that induces potent type 1 interferon (IFN) responses. By directly modifying the STING protein via RONS production, HSV-1 can dampen the IFN response [69]. HSV-1 also depletes the cellular antioxidant glutathione peroxidase 4 (GPX4) during infection. Since GPX4 is an activator of cGAS/STING, its depletion by HSV-1 can prevent the activation of this detection pathway. Without detection of HSV-1 by the cell, viral-induced oxidative stress accumulates, which can be in the form of lipid peroxidation by-products, and further overwhelms cellular machinery that maintains redox balance [70]. This control that HSV-1 holds over antioxidant concentrations can further be imposed through the Nrf2-antioxidant response element (Nrf2-ARE) pathway, which mediates expression of antioxidant genes. Through its ability to modify the Nrf2 transcription factors, HSV-1 can control the redox state in the cell to favor its replication [71].

### 3.2. HSV-1 Maniplates the Cell Environment to Facilitate Viral Gene Expression and Assembly

As a DNA virus, HSV-1 diverts the cellular machinery in the nucleus to replicate its own viral genome and produce new progeny viruses. Once the cell replication machinery is taken over by HSV-1, the virus begins to express its genes in three distinct but overlapping phases: immediate early (IE), early (E), and late (L). All phases of viral gene expression involve viral and cellular factors [9,72]. First, IE genes encode α proteins, which play important roles in enhancing HSV-1 infection by promoting virus replication and dampening antiviral responses in the host cell [50,73,74,75]. The IE phase includes the expression of the E3 ubiquitin ligase ICP0 (also packaged in the virus as a tegument protein), which induces oxidative stress and targets host cell proteins involved in the antiviral defense for degradation [64]. ICP27 is also encoded by IE genes and inhibits cell protein translation alongside tegument protein virion host shutoff (vhs) [63]. IE gene products also initiate a transcriptional cascade, resulting in the expression of downstream E genes that mediate HSV-1 DNA replication [73,76,77,78,79] and L genes that encode protein components used to assemble new viral particles [65,72,73]. L genes also encode ICP34.5 that further modulates the oxidative stress responses in cells by directly binding to Beclin 1, an autophagy-stimulating protein. Through this interaction, ICP34.5 can directly prevent autophagy of the cell, which is often activated by virus-induced oxidative stress as a means to eliminate HSV-1 from the cell [55,80].

During the later events in viral replication, HSV-1 must evade cell detection mechanisms and interfere with signaling to negate anti-HSV-1 immune responses [81]. Interference with antiviral responses is achieved by the manipulation of redox homeostasis in the cell by HSV-1. Specifically, HSV-1 can induce oxidative stress conditions to modify key signaling proteins, resulting in their degradation or loss of function [13,82]. In addition to the ubiquitin activity of ICP0, carbonylation is another type of post-translational modification that can divert cellular proteins from their roles in signal transduction. Carbonylation is described as the introduction of ketones and aldehydes to amino acid side chains. These modifications are recognized as indicators of oxidative stress during viral infections. Proteins with these modifications are inactivated and targeted for degradation in the proteasome. As a result, HSV-1 has adopted carbonylation to target key host cell proteins, such as apoptotic proteins, that would otherwise interrupt productive replication [46,64,83,84].

Cellular mechanisms are also manipulated by HSV-1 to promote virus assembly. Cellular proteins are preferentially degraded over viral proteins by HSV-1 induced expression of virus-induced chaperone-enriched (VICE) domains in the nucleus. In addition, heat shock protein 27 (Hsp27), a molecular chaperone that aids in protein folding, was found to be associated with VICE domains in the nucleus during HSV-1 infection. It is likely that the increased presence of Hsp27 is correlated with promoting viral protein folding in preparation for assembly [64,85].

Following HSV-1 genome replication, the virus begins the transcription of L genes that encode structural viral components required to assemble progeny virions. This assembly process is complex, requiring several viral proteins to form the capsid structure and initiate insertion of the newly replicated genome into the nascent virus particle [86,87,88,89]. Once the genome is packaged into the viral capsid, viral tegument proteins assist the progeny virion in budding from the inner nuclear membrane into the perinuclear space to gain a temporary lipid envelope [87,89]. Virions then de-envelope as they bud into the cytoplasm and are most likely taken up by the trans-Golgi network (TGN) where they are assumed to obtain their tegument protein layers. The TGN is proposed to serve as the site of secondary envelopment, in which virion envelopes derive from vesicles that bud from the TGN to the plasma membrane. The assembled viruses then use the host cell microtubule system for transport to the plasma membrane where the fully assembled virions egress the host cell via exocytosis [90,91,92,93] (Figure 2).

## 4. Latent Infection Induces Oxidative Stress for HSV-1 Persistence in Patients

The alternative outcome of HSV-1 replication is a latent infection. Latency is a state in which virus gene expression and replication are silenced to escape cellular immune detection and clearance, and to allow for long-term persistence of the viral genome in the infected cell. During acute HSV-1 infection, progeny viruses produced by lytic infection of epithelial cells leave the mucosal epithelium, enter the trigeminal nerve, and latently infect the cell bodies of sensory neurons. Since genes associated with replication and viral assembly are expressed to a much lesser extent, patients exhibit no clinical symptoms of HSV-1 infection [50]. The ability of HSV-1 to remain transcriptionally dormant and evade immune system surveillance contributes to viral persistence, allowing lifelong infection with HSV-1 and its continued spread in the general population. While the exact mechanisms by which HSV-1 establishes and maintains latency are not fully understood, evidence suggests HSV-1 can manipulate redox homeostasis within the neurons to promote their survival and continue to escape immune detection [47,82].

Latent infection of neurons begins with mechanisms that parallel those involved in lytic infection but quickly diverges as the virus enters the neuron. Like lytic infection of epithelial cells, initial attachment of the HSV-1 virion to neurons involves interactions between the envelope glycoproteins and cell surface proteoglycans, including HS and nectin-1 [66,94]. Following attachment to these molecules, entry into the neuron occurs via a pH-independent manner that results in the fusion of the virus envelope with the neuronal plasma membrane, a process mediated by the host cell meshwork of filamentous actin [94]. Rather than trafficking via endocytic pathways, the released encapsulated genome uses the neuronal cytoskeleton to initiate retrograde axonal transport from the entry point on the axon. This transport mechanism involves microtubule networks running along the axon to the cell body where latency is established [61,94].

When HSV-1 reaches the cell body of the neuron, its genome rapidly condenses into heterochromatin via epigenetic mechanisms, making genes associated with lytic infection inaccessible to the host transcription machinery. Instead, the condensation of the genome facilitates the expression of a single transcript, the latency-associated transcript (LAT), expressed under a neuron-specific promoter [55,65,95,96,97,98,99,100]. In addition to silencing replication-associated genes [101], LAT expression is responsible for upregulating cellular genes that promote neuron survival and HSV-1 persistence [97,102].

### 4.1. HSV-1 Latency Results in Long-Term Oxidative Damage in Neurons

The nervous system, where HSV-1 resides during latency, is highly susceptible to oxidative stress. In particular, the trigeminal ganglia at the base of the brain are characterized by high oxygen uptake and generally low levels of antioxidants [8,103]. Therefore, during HSV-1 latency, cellular genes induced by the LAT may promote the onset of oxidative stress. Because the brain is rich in polyunsaturated fatty acids, oxidative stress may cause lipid peroxidation reactions, which are known to interfere with cell signaling pathways [15,16]. In fact, HSV-1 infection of neurons is correlated with the release of hydroxynonenal (HNE) and malondialdehyde (MDA), which are by-products of RONS-mediated lipid peroxidation. During HSV-1 establishment of latency, these products can contribute to the downregulation of immune pathways such as NF-κB to negate host innate immune responses and upregulate pathways such as c-Jun N-terminal kinase/mitogen-activated protein kinase (JNK/MAPK) to promote survival of the infected cell [47,104]. Manipulation of these cellular processes contributes to reduced immune visibility and neuron survival, which facilitates the persistence of HSV-1 in the latent stage. To further promote persistent infection, LAT offers protection to the condensed and mostly silent genome of HSV-1 within a circularized episome in the neuron until its eventual reactivation [66,96,105].

Persistence of HSV-1 in the neuron and the associated chronic oxidative stress can result in damage to infected neural tissues. Although latency allows the evasion of HSV-1 from immune clearance, the host still tries to protect itself. Dissemination of HSV-1 to neural tissues can lead to oxidative stress from microglia cells, a common cell type in the nervous system involved in protective immune responses. They are cited as being major cytokine producers and influencing excessive inflammatory responses to HSV-1 infection. Over time, these stressful conditions, in some cases, can lead to serious disease such as encephalitis in the brain, resulting in neuronal death and irreversible brain tissue damage [4,47,106,107,108]. Although the exact mechanisms for disease development are unknown, Alzheimer’s Disease and other neurodegenerative conditions have been associated with HSV-1 infection [6,7,8]. These potential disease outcomes highlight the importance of redox homeostasis and how critical it is in determining the fate of HSV-1 infection and efficiency of the cell to overcome it.

### 4.2. Oxidative Stress in the Reactivation from Latency

Reactivation from latency is a hallmark of HSV-1 infection and occurs periodically throughout the lifetime of an infected patient. This process involves the escape of HSV-1 virions from neurons and re-infection of the innervated epithelial cells within the mucosal epithelium via anterograde transport. Upon re-infection of epithelial cells, the ensuing productive infection of epithelial cells results in the re-appearance of clinical symptoms [2,65,105].

The process of reactivation has long been associated with an individual’s response to external and internal stress stimuli that trigger the escape of HSV-1 from neurons. Regardless of the stimulus, HSV-1 takes advantage of the stress stimuli to escape immune detection in order to re-infect epithelial cells within the mucosal epithelium, resulting in the re-emergence of clinical symptoms [105]. While stimuli involved in cellular stress are known to trigger reactivation, the exact mechanisms by which this process occurs are poorly understood. Overall, reactivation is critical for the lifecycle and persistence of HSV-1. Due to the involvement of oxidative stress conditions in maintaining latency and triggering reactivation, the latently infected neurons are subjected to long-term oxidative damage [8,109].

#### Oxidative Stress as Stimulus in Models for HSV-1 Reactivation

Both in vitro and in vivo models of HSV-1 infection have been used to demonstrate the effects of external stimuli in HSV-1 reactivation from latency. Many of these external stimuli are tied to the induction of oxidative stress responses that push the potential of reactivation of latent HSV-1. Early studies of HSV-1 reactivation focused on the delivery of electrical stimulation of the trigeminal nerve [110]. A commonly used reactivation stimulus is ultraviolet (UV) light, which causes oxidative stress responses in cells through the generation of RONS [111]. High intensities of UV-B light were found to be sufficient to reactivate HSV-1 replication in a coculture system involving HaCaT cells (a neurofibrillary keratinocyte cell line susceptible t1o HSV-1 in vitro), and PC12 neuronal cells (a cell line susceptible to HSV-1 in vitro) [112]. UV light has also been a tool for reactivation in latently infected mice [113]. Relative to mechanism of action, damage to cellular macromolecules by UV light exposure can impact enzyme functions, create lipid peroxidation by-products, and damage nucleic acids [114,115,116]. However, oxidative stress was also shown to aid in reactivation through the administration of sodium arsenite, an inducer of heat shock, and gramicidin D, a toxin that interacts with membrane lipids, both of which are inducers of oxidative stress. Using ICP0-null mutant strains of HSV-1, these agents were capable of reactivating replication in absence of the ICP0 tegument protein, which is implicated in promoting lytic gene expression [117]. Other triggers have been attributed to emotional stress or hormonal imbalances [105,110,118]. This includes adrenergic hormones involved in the body’s stress response such as epinephrine and norepinephrine, which have been shown to reactivate HSV-1 in vivo using a latent rabbit infection model [110]. These studies demonstrate that exposure to harmful stress stimuli can cause the re-emergence of infection and clinical disease via the reactivation of acute HSV-1 infection.

The reactivation of HSV-1 replication can also be affected by the onset of oxidative stress caused by intracellular stimuli. Reductions in the temperature or superinfection with cytomegalovirus (another member of the *Herpesviridae* family known to induce oxidative stress) can activate HSV-1 infection from latently infected cells [119,120,121,122]. Overall. the redox state of cells involved in HSV-1 replication can contribute to reactivation of HSV-1 replication.

## 5. Cellular Mechanisms That Control RONS Concentrations

RONS have dual roles in cells. RONS are naturally produced by the cell and aid in the maintenance of homeostasis. When tightly controlled, RONS are beneficial to the cell, aiding in cell survival, signaling, and immune responses. However, when this control of RONS concentrations is lost, which occurs during HSV-1 infection, RONS can damage cellular macromolecules and dysregulate cellular processes.

To regulate the concentrations of RONS and protect the cell from oxidative damage, cells utilize an antioxidant system composed of enzymatic and non-enzymatic defenses. These defenses are typically used to prevent the accumulation of primary RONS by converting them into less reactive molecules [21,42,44]. For instance, superoxide dismutases (SOD) remove superoxide radicals that are produced during metabolism, generating hydrogen peroxide (another primary RONS and an important signaling molecule) and molecular oxygen, both of which can readily be used by the cell [14,42,123]. Then, to prevent the subsequent accumulation of hydrogen peroxide, catalases convert hydrogen peroxide to water and oxygen [14,41,57]. In addition, glutathione (GSH), in conjunction with other non-enzymatic antioxidants, also controls RONS concentrations in the cell as one of its numerous functions. As a co-substrate for glutathione peroxidase (GPx), GSH limits peroxides through the formation of glutathione disulfide (GSSG), which is then reduced back to GSH, making the balance between GSH and GSSG a useful indicator of the antioxidant capacity in a cell [44,69,124]. Other non-enzymatic scavengers for RONS include vitamins, carotenoids, flavonoids, and melatonin [46] (Figure 3).

### 5.1. Cells Control the Upregulation of RONS in Response to HSV-1

The tightly regulated generation and destruction of RONS in the cell contributes to the antiviral defense mechanisms employed by the cell against HSV-1. In response to HSV-1 infection, cells try to maintain a stable redox environment to prevent the virus from hijacking the cell.

RONS can serve as potent signaling molecules by modifying key cellular proteins involved in signaling pathways and making them integral components of the cellular innate immune system. NOX enzymes, involved in RONS production, influence many toll-like receptor (TLR)-stimulated pathways that can activate or inhibit cellular responses to a pathogen [42]. TLRs are pattern recognition receptors (PRRs) located on the plasma membrane, within endosomes, and in the cytoplasm of cells. TLRs recognize conserved motifs within pathogen structures and upregulate innate immune responses that promote pathogen clearance from a cell. Specifically, TLR2 and TLR4 reside on the plasma membrane of cells, while TLR9 is located in endosomal compartments [42,55,69]. TLR pathogen recognition upregulates innate immune responses that promote pathogen clearance from a cell. Upon stimulation of these TLRs by binding viral components, NOX expression is reported to be upregulated, inducing the production of RONS [42]. Additionally, TLRs promote mitochondrial reactive oxygen species (mROS) generation to aid in immune responses [125]. Lastly, retinoic acid inducible gene 1 (RIG-1)-like receptors, another class of PRRs, have been implicated in their ability to mediate NOX expression by directly sensing foreign DNA and RNA molecules in the cytoplasm [42].

PRRs play active roles in sensing and activation of the host cell immune response, including the generation of RONS. HSV-1 envelope glycoproteins are recognized by PRRs such as TLR2 and αvβ3 integrin on the plasma membrane. Specifically, gH/gL glycoprotein heterodimer was sufficient for this activation and subsequent immune response [126,127]. Intracellularly, endosomal TLR3 and TLR9, located on the ER, have also been shown to recognize HSV-1 structural components and synergize with TLR2 to elicit an inflammatory response and produce RONS [104,128]. In the cell, DNA/RNA sensors such as cGas, RIG-1, and γ-interferon inducible protein 16 (IFNI16) sense HSV-1 during the replication cycle to trigger RONS production and promote innate immune responses [104,127,128,129].

RONS produced during viral infection can lead to oxidative damage to cellular macromolecules, which can prompt immune responses toward the invading virus. Mitochondrial DNA (mtDNA) is a nucleic acid susceptible to RONS-mediated damage from virus-induced oxidative stress due to the lack of DNA repair mechanisms present in the mitochondria. DNA packing proteins that bind the mtDNA to maintain its structure and control the accessibility of genes can also be subjected to oxidative damage by RONS. Overall, damage to mtDNA serves as an antiviral sensor during infection and promotes the stimulation of innate immune responses [130] (Figure 4).

### 5.2. Cellular RONS Mediate Innate Immune Responses against HSV-1

Following recognition of HSV-1 by innate sensors, the cell is prompted to create an antiviral environment to prevent efficient viral replication and assembly. Activation of the NF-κB pathway, known to be influenced by RONS, results in the production of proinflammatory mediators, primarily type 1 IFN, which promote inflammation and T cell responses. These molecules are important mediators in antiviral defenses [126,127,130,131]. While not a direct inducer of NF-κB, micromolar concentrations of hydrogen peroxide are shown to be sufficient for NF-κB activation and function [132]. In the absence of stimulation, NF-κB resides in the cytoplasm bound to the IκBα inhibitor to prevent its translocation into the nucleus. The dynein motor complex protein, LC8, is a protein that can bind and modulate the activity of the IκBα inhibitor bound to NF-κB. Following activation of NF-κB by TNF-α, subsequent RONS production was found to oxidize LC8, releasing it from IκBα, which is then subjected to phosphorylation and ubiquitination [133]. As a result, NF-κB is released, allowing the transcription factor to translocate into the nucleus and promote expression of its target genes. NF-κB responsive genes are known to induce proinflammatory responses including the production of cytokines [15,132,134]. Thioredoxin, a group of redox proteins with a known role in signaling, was also found to modulate NF-κB activation through increased DNA binding activity in bone marrow dendritic cells [135]. Additionally, NF-κB knockout experiments demonstrated that HSV-1 replication and production of viral-induced RONS increased in the absence of NF-κB, highlighting the efficiency of the pathway as an immediate immune response toward HSV-1 while also implicating its susceptibility to redox signaling [81]. Additionally, NF-κB and Nrf2-Keap1 pathways have been shown to counteract each other by competing for the same co-activator during transcription. This is a mechanism used by the cell to regulate these pathways and to control redox levels and inflammatory responses to maintain cellular homeostasis [136,137].

As HSV-1 infection progresses, accumulation of RONS can result in the activation of inflammasomes [127] and autophagy [80] within the infected cell. Intracellular accumulation of RONS in response to HSV-1 infection activates the NOD-, LRR- and pyrin domain-containing protein 3 (NLRP3) inflammasome, resulting in the secretion of proinflammatory cytokines interleukin (IL)-1 β and IL-18, along with caspase-8 [42,125]. Once PRRs within the infected cell recognize HSV-1 components, immune signaling cascades are activated to promote the expression of proinflammatory mediators to attract innate immune cells. These recruited cell types mediate the destruction of pathogens during phagocytosis by phagocytes (e.g., neutrophils and macrophages). These cell types are capable of engulfing viruses or viral components into intracellular phagosomes. Within these phagosomes, oxidative stress conditions, via upregulation of RONS generation, are created to mediate destruction of the phagocytosed pathogen. To generate sufficient quantities of RONS for this function, mitochondria are recruited to the phagosome through signaling cascades to produce mROS and supply electrons in the form of NADPH [138]. As a result, NOX2 can generate superoxide radicals in large quantities, which then dismutate into hydrogen peroxide. Hydrogen peroxide is then consumed by phagosomal myeloperoxidase (MPO) to produce hypochlorous acid, a secondary RONS, through secondary reactions. Other reactive species, such as hydroxyl radicals and singlet oxygen, are also involved in this process [139,140]. This process is commonly referred to as a respiratory burst, known to mediate destruction to the pathogen structure and activate additional signaling cascades that can promote the pathogen’s clearance [22,42,140].

### 5.3. Evidence of RONS Involvement in Adaptive Immune Response against HSV-1

Immune-associated RONS also have roles in the induction of adaptive immune responses. As a novel sensor for HSV-1, TLR3 is linked to the induction of antigen presentation by dendritic cells, which are involved in the induction of a robust CD8+ T cell response against HSV-1. TLR3-deficient mice demonstrate impaired HSV-1-specific CD8+ T cell responses and loss of control over HSV-1 infection in epithelial cells [127]. TLR3 is a sensor that can become activated by oxidative stress induced during viral infections [141]. Therefore, its role in promoting CD8+ T cell responses could be influenced by RONS. Nitric oxide was also observed to modulate the adaptive immune response by regulating the proliferation of lymphocytes in response to HSV-1, while also recruiting immune cells for antigen presentation [39]. These findings indicate that RONS can facilitate adaptive antiviral responses and are critical mediators of the cell-based immune response against HSV-1.

## 6. Examination of RONS as Antiviral Agents

The potent antiviral capabilities of RONS have garnered interest in the development of RONS cocktails as antiviral therapies. RONS have demonstrated roles in the cellular antiviral response, and some species are currently utilized as the basis of disinfectant agents for the decontamination of surfaces.

By itself, ozone inactivates a multitude of viruses. For example, ozone application to cell-free poliovirus-1, which is a single-stranded RNA virus, was shown to modify the protein sequences within the viral capsid, resulting in the impairment of viral adsorption. Furthermore, ozone damaged the viral RNA genome, leading to its inactivation [142]. Similarly, a 3-h application of ozone to cell-free HSV-1 resulted in 90% inhibition of viral infection. Not only did ozone treatment reduce viral infectivity, it was also shown to induce cytokine expression in the infected cell that promoted an innate immune response [143]. Ozone can enter the liquid phase and interact with other RONS such as nitrogen dioxide to further enhance its antiviral effect. Specifically, the concurrent presence of ozone and nitrogen dioxide in liquid media coincided with the enhanced generation of secondary RONS, such as dinitrogen pentoxide [144]. As a secondary species, dinitrogen pentoxide can rapidly accumulate and contribute to the oxidative damage of viruses. Like ozone, dinitrogen pentoxide acts as an antiviral agent in plants by decreasing viral lesions and inducing antiviral immunity in plants [145].

As a common disinfectant, hydrogen peroxide is known to be a powerful antimicrobial agent against viruses and microorganisms. Hydrogen peroxide has antiviral effects against both cell-free and intracellular viruses. Specifically, hydrogen peroxide is virucidal against many viruses spread through contaminated surfaces in healthcare, veterinary, and public facilities. These include feline calicivirus (FCV), transmissible gastroenteritis coronavirus (TGEV), avian influenza virus (AIV), and swine influenza virus (SwIV) [146]. Additionally, hydrogen peroxide was effective for decontamination of human severe acute respiratory syndrome-coronavirus-2 (SARS-CoV-2), with enhanced viral inactivation under acidic conditions through the co-presence of acidic compounds [147]. In a study involving the surface decontamination of influenza virus, a 2-min exposure to hydrogen peroxide vapor resulted in 99% viral inactivation [148]. The virucidal action of hydrogen peroxide was also reported to increase with the co-application of other RONS like sodium nitrite against FCV. This was mostly mediated through the formation of peroxynitrite, a secondary RONS. The activity of peroxynitrite was confirmed with a reduction in the pH of the exposed medium and reduced virucidal activity following the administration of ascorbic acid, a scavenger for peroxynitrite [149]. Hydrogen peroxide can also result in the formation of hypochlorous acid, a reactive chlorinated species, in a reaction catalyzed by MPO. Hypochlorous acid is commonly produced in leukocytes to cause oxidative stress conditions during phagocytosis [150]. Hypochlorous acid is also used as a disinfectant and is shown to inactivate viruses [151]. Some studies have also investigated hydrogen peroxide as a possible vaccine supplement, given its potent signaling capabilities. Interestingly, in a hydrogen peroxide-inactivated west nile virus (WNV) vaccine, specific CD8+ T cell and antibody-mediated responses were observed, resulting in immune memory upon re-infection with WNV [152].

Singlet oxygen is a short-lived RONS known to directly oxidize lipids containing carbon double bonds. Singlet oxygen is most effective against enveloped viruses (viruses enclosed by lipid envelopes). Singlet oxygen inactivates viruses by disrupting the viral envelope, which compromises its capacity to mediate entry into a target cell. In a singlet oxygen inactivated pseudorabies virus (PRV) vaccine, oxidized lipids within the virion structure also elicited a strong antibody-mediated response, while reducing PRV infectivity [153]. Singlet oxygen has also been reported to modify key amino acids in protein structures within the FCV capsid. This includes amino acids that contain double bonds in their side chains and are susceptible to disulfide bond formation [149]. As a therapy, singlet oxygen generation is typically seen during photodynamic therapy (PDT) to inactivate viruses like SARS-CoV-2. However, it must be delivered to target cells immediately after generation for maximum therapeutic effect [154].

Endogenously, nitric oxide is a primary RONS involved in cellular antiviral defense. Following viral sensing by cellular PRRs, immune signaling pathways (e.g., NF-κB) are activated to promote the expression of proinflammatory cytokines that enhance the recruitment of innate immune cells to the site of viral infection. Additionally, immune signaling pathways involved in antiviral defense can further upregulate the expression of iNOS to promote nitric oxide generation [155]. Nitric oxide also has a critical role in modulating immune responses during early HSV-1 infection as inhibition of nitric oxide resulted in higher levels of HSV-1 replication and increased disease pathology in brain tissues [156]. Given its role in the antiviral defense, some studies have explored the use of exogenous nitric oxide as an antiviral therapy. Recently, this activity was demonstrated through the application of nitric oxide plasma activated water (NO-PAW), which allowed the delivery of gaseous nitric oxide into water, to samples of SARS-CoV-2 infected cells. Treatment of infected cells with NO-PAW resulted in the suppression of spike protein expression, the entry protein for SARS-CoV-2, along with other key viral proteins. In addition, NO-PAW was also able to upregulate the antiviral gene response in cells [157].

## 7. NTP as a Method for Controlled RONS Delivery

Although RONS are produced by and used by the cell in its antiviral defense, these species are often not produced in quantities sufficient to effectively control viral infections. Therefore, strategies that boost RONS and their effects have been explored as stand-alone antiviral therapies against viral infections. Many of the potentially therapeutic RONS cannot be produced artificially due to their stringent generation requirements and short half-lives. Furthermore, clinical approaches that use chemistry-based methods to generate RONS are often limited by the short half-lives of chemically active products and the identification of effective means to deliver them. These limitations can be addressed in therapeutic approaches involving the application of non-thermal plasma (NTP).

The field of plasma medicine is defined by explorations of NTP as a tool for the controllable delivery of RONS to biological targets. Also referred to as cold atmospheric plasma (CAP), gas plasma, or low temperature plasma (LTP), NTP has emerged as a relatively safe therapeutic tool with applications in wound healing, cancer, and infectious disease [158]. As the fourth state of matter, NTP is defined as a partially ionized gas composed of chemical, electrical, radiative, and thermal components (Figure 5). During the generation of NTP, the ionized gas produced at ambient temperature and pressure, with the application of electric fields, can excite and ionize electrons to produce RONS [158,159,160,161]. The quantities of RONS produced are regulated by changing applied electric fields, voltage, frequency, duty cycle, time, and distance of application, depending on the device. With their controllable delivery via NTP, these highly reactive species have the ability to modify macromolecular structures, induce intracellular signaling cascades, cause immunogenic cell death (ICD), and induce immune responses in a biological target [162]. While NTP is composed of multiple components, the ability to generate multiple species of RONS (some of which cannot be produced naturally by the cell or synthesized by chemical-based approaches) is critical to its ability to cause biological effects, including the inactivation of many types of viruses [163].

### 7.1. NTP Adds an Additional Layer of Oxidative Stress to Control HSV-1 Infection

Redox homeostasis is an important determinant in the survival of cells. During viral infections, this homeostasis is challenged by oxidative stress induced by the cell to protect itself and by the virus to promote its replication and pathogenesis. Therefore, the redox state of the cell during a viral infection determines the likelihood of the cell being able to control the viral infection. Given the effective hijacking mechanisms of viruses, infections usually overwhelm the cell and allow the virus to utilize its machinery for its replication, which downregulates cellular DNA replication. With NTP and its controllable delivery of antiviral RONS, this redox state can be shifted back in the cell’s favor to promote interference with virus replication.

While RONS are important mediators of antiviral activity, other components of NTP also contribute to its antiviral activity. By themselves, electric fields and UV radiation can kill microorganisms and can act as stand-alone therapies for treating infections [164,165,166,167]. UV light also induces oxidative stress [111,168,169] and can promote the generation and delivery of RONS to an infected cell. Additionally, cells exposed to electric fields and UV light induce the generation of RONS, activating key cellular signaling pathways in response to macromolecular damage and apoptosis [160,170,171]. Pulsed electric fields, which cause membrane permeabilization and are the basis of electroporation, also induce both extracellular and intracellular RONS in cells [172]. Although NTP can sometimes cause slightly elevated temperatures in biological targets, the negligible temperature increases caused by NTP application have little to no impact on virus inactivation [173]. It is important to note that no single component is produced in enough quantities to have a stand-alone effect. It is hypothesized that the different components of NTP work synergistically to produce the observed antiviral response. Therefore these subtherapeutic amounts of each component, working together, make NTP a unique tool for inactivating viruses [160].

### 7.2. Specialized Devices Allow Controllable Delivery of RONS

NTP is generated and delivered to a biological sample through specialized devices. In research and clinical settings, two main types of devices are used. Dielectric barrier discharges (DBD) devices allow the direct application of NTP to samples. These devices consist of at least one electrode encased in a dielectric material to which a high voltage is applied. The counter electrode could be the biological substrate as shown in Figure 5 for the example of a so-called floating DBD. NTP-generated RONS are directly deposited onto samples across small gap distances between the sample and the DBD electrode [158,174,175]. In addition to RONS, the direct interaction between the plasma and the sample delivers ions, UV photons, and high electric fields to the biological sample. In contrast, plasma jets are designed with a tube-like configuration that houses an electrode. Unlike DBD devices, plasma jets generate NTP at a larger distance from the target. The components of NTP are directed onto the target by the flow of typical inert gases such as helium or argon [158,176]. Plasma jets can operate in two modes: with and without direct contact of the plasma to the biological sample. When there is direct contact with the biological substrate, the interactions with a plasma jet are similar to interactions with DBD plasma as shown in Figure 5. When there is no direct interaction of plasma with the substrate, the substrate is not subjected to high electric fields and ionic species and biological interactions are mainly due to RONS. Although both types of NTP devices generate RONS, the composition of RONS delivered to the sample varies considerably between devices. Furthermore, the composition and quantity of NTP-generated RONS can be altered in a device-specific manner through adjustments in voltage, frequency, exposure time, electrode distance to the target, and type of working gas. The parameters that mediate the delivery of RONS are specific to the type and design of the device and application modality [160,163].

The composition of the RONS generated is also highly dependent on the power supply utilized to generate the NTP, as well as the sample being exposed. In the gas phase, the working gas and the parameters that result in NTP generation can influence the composition of RONS produced. As these species are delivered to a target, the interactions with the liquid-phase (typically cell culture medium or interstitial fluids in in vitro or in vivo applications) can result in the formation of secondary species [177]. Among the various RONS that can be produced, ozone (O_3_), singlet oxygen (^1^O_2_), hydroxy radicals (OH), hydrogen peroxide (H_2_O_2_), nitric oxide (NO), nitrogen dioxide (NO_2_^−^), peroxynitrite (ONOO^−^), and dinitrogen pentoxide (N_2_O_5_) have been implicated in pathogen inactivation [149,163]. The capability to generate and deliver these RONS controllably makes NTP a potential antiviral agent that can be used effectively to treat infections by viruses such as HSV-1.

## 8. NTP as a Virucidal and Antiviral Agent

The antiviral activity of NTP-associated RONS has been clearly documented and summarized by Mohamed et al. [163]. Due to NTP’s ability to inactivate viruses and reduce their infectivity, NTP can be a promising therapeutic preventative measure for infections by human and non-human viral pathogens. NTP also has the ability to inactivate the plant virus, Potato Virus Y, which is a common drinking water contaminant. Studies involving NTP and this virus demonstrated NTP-mediated viral inactivation and proved NTP to be an environmental-friendly and safe decontamination tool for use in irrigation systems [178]. NTP was also shown to be an effective decontamination tool for foodborne viruses such as norovirus (NoV), a single-stranded nonenveloped virus found in fecal-contaminated produce and drinking fluids. Not only was its inactivation found to be consistent with increasing exposure times to NTP, but RONS were speculated to be the dominant effectors in its antiviral mechanism [179].

The animal virus FCV, a surrogate for human NoV, has also been the focus of research to test the antiviral efficacy of NTP. Using an argon plasma jet, FCV inactivation was more apparent with shorter exposure distances and increasing power. A correlation between NTP exposure and the oxidation of FCV viral capsid mediated by the presence of NTP-generated RONS was suggested. Specifically, modifications of the capsid and viral inactivation were proposed to be influenced by the presence of RNS, ozone, singlet oxygen, and the formation of peroxynitrite [173]. The antiviral effect of NTP against FCV has been shown using a variety of plasma devices and operational conditions, including remote plasma treatments with the effluent of a 2-dimensional (2D)-DBD plasma, which delivered mainly long-lived RONS to the virus. When NTP was applied directly, FCV inactivation could be induced by the actions of ozone in the gas-phase. In contrast, when NTP exposure was indirect, through application of liquid enriched in NTP RONS, NOx species were the dominant effectors. Additionally, pH changes secondary to peroxynitrite formation played a role in FCV inactivation [180]. Other studies that focused on FCV inactivation also highlighted ozone and peroxynitrite as the key antiviral agents in NTP function by producing oxidative damage to key amino acid residues within the viral capsid of FCV [144,181].

Few studies have investigated NTP as a potential therapeutic alternative for viruses that infect humans and cause chronic disease. Hepatitis B virus (HBV), a viral pathogen associated with a chronic liver disease, is spread through contact with bodily fluids. NTP, generated by a DBD device, was applied to blood samples infected with HBV. Key HBV antigens were susceptible to RONS-mediated damage following NTP exposure, leading to inactivation of the virus that increased with longer durations of exposure to NTP [182]. The antiviral effect of NTP was also studied using cells infected with HIV-1. NTP exposure of HIV-1-infected cells reduced HIV-1 infectivity, and impaired virus-cell fusion and viral assembly. Additionally, it was speculated that NTP exposure induced cellular factors that promoted an antiviral environment in macrophages, preventing further viral entry [183]. Similarly, viral entry mechanisms were abolished by NTP application to SARS-CoV-2. Specifically, the viral spike protein was found to be altered in its conformation and impaired in its ability to bind to cellular receptors. NTP exposure decreased the infectivity of SARS-CoV-2, and even resulted in the disruption of cell membranes which led to oxidative damage of viral RNA inside the cell [184].

The antiviral effect of NTP on HSV-1 was investigated in a model for herpes keratitis, in which explanted HSV-1-infected human cornea cells were indirectly exposed to NTP. In these experiments, cell culture medium was exposed to NTP and then applied to the HSV-1-infected cells, resulting in an 80% reduction in viral infectivity and increased antiviral activity with longer durations of NTP exposure. Importantly, minimal host cell cytotoxicity was observed [185]. An increase in 8-oxodeoxyguanosine (8-OHdg), a marker for oxidative DNA damage correlated with increased virus inactivation without damage to cells [186].

## 9. NTP as a Therapeutic Alternative for HSV-1 Infection

Oral, ocular, and genital lesions are considered hallmarks for HSV-1 infection and appear when the virus is actively replicating in mucosal epithelial cells [2,3]. In an envisioned NTP-based therapeutic approach, NTP will be applied directly to lesions produced by active HSV-1 infection. The application of NTP to HSV-1 lesions will deliver NTP effectors to the local area of the epithelium that includes cell-free virus, cells undergoing productive infection, and uninfected cells that may be targets for HSV-1 infection. Direct application of NTP to these lesions will likely reduce virus infectivity through oxidative damage to virus components, including envelope proteins and the DNA genome. NTP may also indirectly have an antiviral effect by altering a host cell’s capacity to support productive infection and preventing infection of uninfected cells by altering cell surface molecules that participate in viral binding and entry.

### 9.1. Direct Effects of NTP-Generated RONS on HSV-1 Infection

Studies to date have focused on the short-term antiviral effects of NTP using cell-free HSV-1 and HSV-1 infected cells, with changes in viral infectivity as the measure of the antiviral effect of NTP. The in vitro antiviral effect of NTP on HSV-1 infection was demonstrated through the application of NTP-treated cell culture medium to HSV-1-infected cells (indirect NTP) [185]. There are multiple mechanisms through which NTP could directly affect HSV-1.

NTP is proposed to act as an antiviral agent through the controllable delivery of RONS, which are known to interact and modify macromolecular structures. Proteins, lipids, and nucleic acids in enveloped viruses such as HSV-1 are likely susceptible to RONS-mediated damage. For example, cysteine, a sulfur-containing amino acid, is prone to disulfide bond modification by RONS [187]. Additionally, MPO, an enzyme produced by cells during phagocytosis, can convert hydrogen peroxides into ions that have been implicated in the oxidation of tyrosine residues [20,188]. Given the abundance of proteins within the HSV-1 virion structure (e.g., envelope, capsid, and tegument proteins), RONS generated by NTP likely impair HSV-1 infectivity through protein modification. RONS can also affect carbon double bonds in lipid-based structures, resulting in lipid peroxidation. The viral envelope, which is important for protecting HSV-1 from the extracellular environment and for mediating fusion with target cells, can be disrupted by oxidative stress conditions through lipid modification [153]. Lastly, nucleic acids are prime targets for RONS-mediated damage. Specifically, RONS-induced modifications are typically observed within the sugar backbone of RNA and DNA molecules. Base pairing can also be disrupted by RONS, leading to the introduction of mutations in the viral genome that can impair HSV-1 infectivity [20]. Although not proven, some studies have alluded to RONS-mediated damage of viral macromolecules as the mechanism underlying the antiviral activity of NTP. This conclusion was supported by measurements of 8-OHdg, which is a marker for oxidative DNA damage. Following NTP exposure of HSV-1-infected ocular cells, increases in 8-OHdg correlated with the inactivation of HSV-1 [186].

Given the complexity of the HSV-1 structure, many viral components can be targets for damage caused by NTP RONS. As previously described, this damage will take the form of modifications to viral macromolecular components, leading to impairment of their functions. Due to the location of the envelope and envelope proteins on the exterior of the virus particle, reductions in virus infectivity attributable to NTP exposure are most likely to result from oxidative damage to the viral glycoproteins that mediate viral entry and modifications of lipids within the viral envelope that protect the encapsulated genome. Due to its chemical stability and hydrophilicity, hydrogen peroxide is an example of a RONS that can penetrate membranes [189]. As a result, hydrogen peroxide may have the ability to pass through the HSV-1 lipid envelope and gain access to the interior of the HSV-1 virion structure, causing oxidative damage to viral proteins and DNA contained within. These effects may also contribute to reductions in infectivity.

### 9.2. Effects of NTP-Induced Oxidative Stress on HSV-1 Replication

Redox control in a cell is important for cellular homeostasis and preventing damaging oxidative stress conditions. In the context of HSV-1, there is a clear correlation between infection and the induction of oxidative stress [85]. Part of this stress response is created by the cell. RONS are often produced to trigger intracellular signaling cascades that result in the promotion of an antiviral environment that is unfavorable for HSV-1 infection [104,127,128]. Meanwhile, HSV-1 can induce its own oxidative stress response that overcomes the cell’s immune defense and enhances its own pathogenesis. This often involves the degradation of host proteins, recruitment of cellular machinery for its replication, and interference with signaling pathways through the modification of macromolecules [44,47,57,69]. Overall, HSV-1-induced oxidative stress aims to impair the cell’s ability to promote its immune clearance, allowing it to replicate or persist in a dormant state within the infected cell.

Induction of oxidative stress by HSV-1 can be observed through the recruitment of VICE domains, the localization of heat shock proteins, and the activation of the unfolded protein response (UPR) in the ER. To promote the assembly of proteins for viral replication, HSV-1 induces ER stress to target cellular proteins for degradation to promote the proper assembly of virions [64,85]. This results in the accumulation of oxidized proteins and causes further dampening of the immune response towards HSV-1. In a model for diet-induced obesity, proteins associated with ER stress and the activation of the UPR were downregulated in NTP-exposed adipocytes, both in vitro and in vivo, suggesting that NTP may inhibit ER stress [190]. This ability of NTP to inhibit ER stress may allow the restoration of redox homeostasis in cells subjected to HSV-1 infection. NTP is also implicated in the increased expression of antioxidants that detoxify accumulating RONS concentrations, contributing to efforts in normalizing cellular redox levels. NTP exposure of HaCaT cells upregulated Nrf2 signaling, a transcription factor in the Nrf2-Keap1 pathway involved in the transcription of antioxidant genes as early as 20 s post-exposure to NTP and up to 24 h later [191,192]. Additionally, hydrogen peroxide, a key component of NTP, partially decreases Keap1, a negative regulator for Nrf2 expression [191]. Furthermore, NTP increases GSH levels, another antioxidant important in hydrogen peroxide neutralization [193]. This suggests that application of NTP has the potential to promote regulation of RONS concentrations in the cell, compromising the ability of HSV-1 to manipulate oxidative stress in the host cell.

Although NTP has a demonstrated antiviral effect on HSV-1, the mechanism by which RONS generated by the application of NTP to infected cells contributes to the antiviral effect of NTP is unclear. Based on reports cited above, it may be speculated that NTP exposure of HSV-1-infected cells could potentially stabilize the redox balance in a cell, diminishing oxidative stress mechanisms that favor replication.

## 10. NTP as an Immunomodulatory Agent for Treatment of HSV-1 Infection

In addition to its antiviral activity, NTP has potential as an immunotherapy for HSV-1 infection. RONS, as one of the main effectors of NTP, have demonstrated immunomodulatory effects on cancer cells, inducing immunogenic cell death (ICD), characterized by cell death that elicits an immune response [194,195]. The induction of ICD is normally accompanied by the display of stress-associated molecular patterns (SAMPs) on the cell surface [194,195,196,197,198,199]. These include calreticulin (CRT) and molecular chaperones such as Hsp90. In particular, Hsp90 was cleaved following exposure to NTP, and kinases that aid in its normal function were degraded [197]. Additionally, there is release of other molecules including ATP, high mobility group box protein 1 (HMGB1), and proinflammatory cytokines in response to NTP exposure. The emission of these SAMPs enhanced the functions of innate immune cells, including antigen presentation, facilitated phagocytosis, and promoted an inflammatory response within tumor environments [194,195,196,198]. This resulted in T helper cell activation as evidence of stimulated adaptive immune responses [200].

Literature regarding immunomodulatory changes, or SAMPs, caused by NTP in virus-infected cells is sparse. Our work in this area demonstrated the immunomodulatory effect of NTP in the context of HIV-1 infection using J-Lat cells. These cells, which contain an integrated latent HIV-1 genome that is not capable of supporting multiple rounds of replication, serve as a model for latently-infected T lymphocytes. NTP exposure of J-Lat cells resulted in emission of molecules, characteristic of SAMPs that are typically associated with ICD in tumor cells. Furthermore, there was upregulated phagocytosis by antigen presenting cells, suggesting an adjuvant effect of NTP. There was also evidence of neoepitope generation that may increase the breadth of adaptive immune responses [201]. Similar mechanisms may be activated in HSV-1-infected cells that trigger signaling pathways involved in the cellular immune defense [162].

### Enhancement of Anti-HSV-1 Host Immune Responses by NTP

Innate and adaptive immune responses play integral roles in the host defense against HSV-1 infection. As previously mentioned, TLRs recognize viral components within the HSV-1 virion on the plasma membrane and within the cytoplasm [126,127,128,202]. Upon sensing HSV-1 components, TLRs induce immune signaling pathways that promote the transcription of antiviral mediators by the infected cell. These mediators include type 1 IFNs and other proinflammatory cytokines that are typically induced during viral infections. Type 1 IFNs, in particular, induce the expression of interferon stimulated genes (ISG), the products of which attract patrolling innate immune cells to the site of acute HSV-1 infection [13,55,63,126,128]. This trafficking of immune cells (e.g., neutrophils, macrophages, and dendritic cells) elicits their effector innate functions, which include phagocytosis and subsequent antigen presentation to adaptive immune cells (e.g., T cells, B cells) [203,204,205]. As a result, both T cells and B cells become activated, stimulating specific adaptive responses toward HSV-1. Specifically, CD8+ T cells, which we propose will be the main mediators of the NTP-stimulated immune response, can secrete antiviral cytokines toward invading pathogens and directly kill infected cells to inhibit viral replication [205]. Additionally, activation of B cells promotes the production and secretion of antibodies against HSV-1 antigens to prevent spread to neighboring susceptible cells [206]. To evade these immune responses, HSV-1 may escape the mucosal epithelium to establish latency in neurons. By silencing its genome expression, HSV-1 avoids sensing by the host immune system [4]. With the application of NTP to HSV-1 lesions, these responses can be stimulated or enhanced, shortening the course of acute infection and forcing HSV-1 into viral latency. Due to this shortened replication cycle in the epithelium from HSV-1′s evasion tactics and the stimulated adaptive immune response, fewer virions will be produced that can travel into the nervous system, decreasing the number of latently infected cells. Over time, this can decrease the frequency and number of reactivation events leading to the recurrence of acute replication or viral reactivation.

HSV-1 has developed many ways to overcome the cellular sensing and immune clearance tactics to persist in its host. As previously mentioned, this includes the manipulation of the cell’s redox state which can influence immune signaling and responses toward HSV-1. Application of NTP may be effective in overcoming viral evasion strategies through the generation of NTP-associated RONS that will enhance and stimulate more robust antiviral responses in the cell. Signaling pathways, such as NF-κB, which induce type 1 IFN responses, are upregulated by NTP exposure [104,126,127,128]. In addition, the application of NTP allows the emission of SAMPs on the cell surface that recruit innate immune cells [201]. These changes can lead to the aforementioned cellular innate and adaptive immune responses directed against HSV-1. The presence of NTP-generated RONS overcomes HSV-1′s mechanisms of suppressing host cell responses. By controlling the cellular redox environment, NTP could prove to be an effective immunotherapy against HSV-1 infection.

## 11. Overlapping Roles for Oxidative Stress in Treating HSV-1 Infection with NTP

The use of NTP as a treatment for HSV-1 infection will require an integrated understanding of the roles played by oxidative stress during infection and treatment. Oxidative stress has multiple functions and effects in immune responses mounted by cells against infection, in HSV-1 replication in host cells, and in cellular and viral responses to NTP application (Figure 6). Some of the effects of RONS and modulated oxidative stress are unique to immunomodulation, infection, or NTP application. However, some roles are common to two or all three aspects of a putative treatment. Overlapping roles for RONS and oxidative stress may be advantageous or detrimental to treatment effectiveness. For example, the boost in cellular RONS by NTP application to an infected cell may augment RONS-mediated cellular mechanisms already promoting effective innate and adaptive immune responses to infection. On the other hand, oxidative damage to macromolecules and organelles during productive HSV-1 infection may be further increased by NTP application. The application of an NTP-based treatment of HSV-1 infection (or other infections or diseases) will need to be conducted at a dose that comprehensively considers the beneficial and detrimental effects of RONS and oxidative stress.

Once considerations of NTP dose are satisfied, the potential net effects of immunomodulation induced by NTP application to an HSV-1-associated lesion are the promotion of more effective innate antiviral responses in infected cells, induction of innate protective responses in nearby uninfected cells, and recruitment of immune cells that will participate in a more effective adaptive response to infection (Figure 7). By controlling the cellular redox environment and modulating both innate and adaptive responses against HSV-1, NTP could prove to be an effective immunotherapy against HSV-1 infection.

## 12. Conclusions

HSV-1 continues to be a global health concern, with high infection rates worldwide and its ability to cause lifelong infection. Although the current antiviral therapies are effective in moderately reducing the severity of symptoms associated with acute infection, they are ineffective with respect to curing a patient after primary HSV-1 infectious due to their inability to address viral latency. Therefore, HSV-1 persists in its host, risking dissemination of the virus to other organs and causing other acute and chronic diseases.

For HSV-1, the manipulation of redox homeostasis is an evasion strategy to overcome immune clearance from the cell and to craft a cellular environment favorable for viral replication. On the other hand, maintenance of redox homeostasis by the cell is crucial in controlling the immune response directed toward HSV-1 and preventing damaging oxidative damage by RONS. Therefore, control over the redox balance within an infected cell is one of the key factors involved in determining the outcome of HSV-1 infection. Given the antiviral properties of RONS in the cell and in commercial disinfectant agents, RONS have the potential to be harnessed as antiviral therapies.

NTP technology is an inexpensive, innovative method of producing RONS responsible for inducing desired biological effects. It has the potential to act as a multi-faceted therapy effective against HSV-1 disease. Decreases in HSV-1 infectivity attributed to the direct antiviral effect of NTP on cell-free virus will decrease the overall viral burden in the lesion, thereby decreasing the clinical symptoms of infection and accelerate resolution of the lesion. An added benefit of reduced titers in the lesion will be a smaller pool of latently infected neurons that serve as reservoirs of long-term infection. The immunomodulatory effects of NTP exposure will augment local innate antiviral effects and boost adaptive immune responses involving HSV-1-specific CD8+ T lymphocytes. The combined antiviral and immunomodulatory activities of NTP are hypothesized to provide short-term relief for acute infection as well as long-term immunological control over reactivation from latently infected neurons, as indicated by reductions in or full control of recurrent lesions in HSV-1-infected individuals.

Of course, the potential of an NTP-based therapy must be fully explored along a developmental path that leads to clinical use. Future studies of the use of NTP as a therapy for HSV-1 infection will include examinations of how NTP can be used to treat lesions, a common manifestation of human HSV-1 infection. Using a preclinical murine model of HSV-1 infection, the antiviral and immunomodulatory effects of NTP on HSV-1 infection will be examined in the context of innate and adaptive, HSV-1-specific immune responses. These models will provide the insights necessary to develop this novel therapeutic as a non-invasive, pain-free, and effective clinical approach against HSV-1 infection.

## Figures and Tables

**Figure 1 ijms-24-04673-f001:**
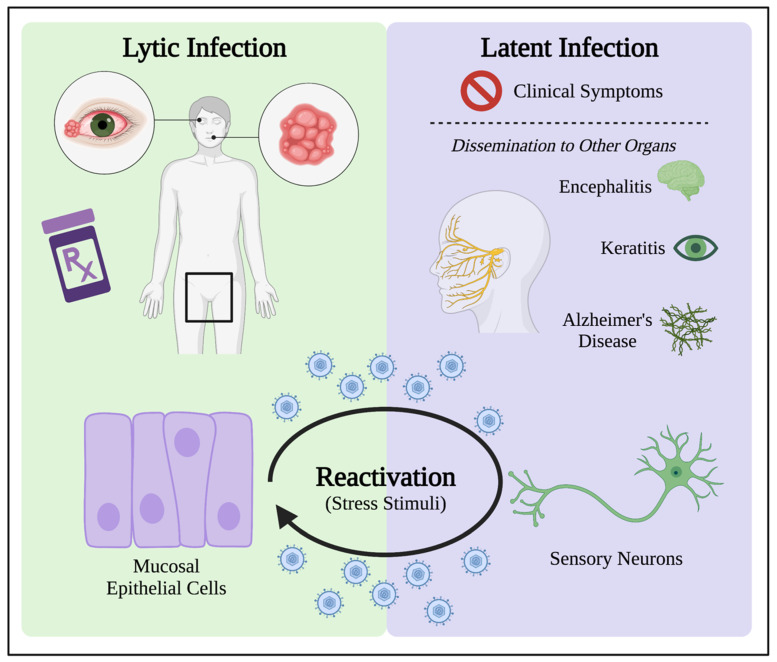
The HSV-1 lifecycle is characterized by the cycle between lytic infection and latent infection over the course of a patient’s life. Infection with HSV-1 is known for causing cold sores that can appear around the mouth, eyes, and genitalia. These clinical manifestations are characteristic of lytic infection, when HSV-1 is actively replicating and producing new viruses in mucosal epithelial cells in these areas. It is at this point in infection where antiviral therapies are administered to alleviate symptoms and shorten the course of virus infection. As clinical symptoms disappear, HSV-1 virions travel to nearby nerves and establish viral latency in peripheral sensory neurons. Latent infection is characterized by negligible viral gene expression, which prevents detection by the host immune system. During the life of the infected individual, HSV-1 lytic infection can become reactivated by stress stimuli. Reactivation can lead to the recurrence of lesions caused by productive, lytic infection, as well as an increased risk of more serious disease.

**Figure 2 ijms-24-04673-f002:**
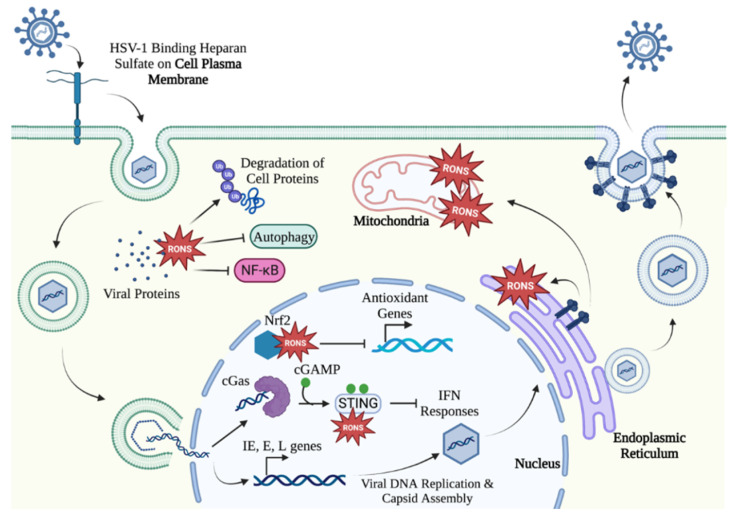
During infection of an epithelial cell, HSV-1 depends on the cell for endosome trafficking and its replication machinery. Due to this dependence, HSV-1 infection is invasive and must continuously avoid immune clearance from the cell. To overcome this, HSV-1 manipulates the cellular redox environment by upregulating oxidative stress responses to overwhelm the cell’s capacity for balancing redox, crafting an environment that promotes HSV-1 replication. This allows the virus to avoid sensing by immune receptors and the subsequent innate responses directed against HSV-1.

**Figure 3 ijms-24-04673-f003:**
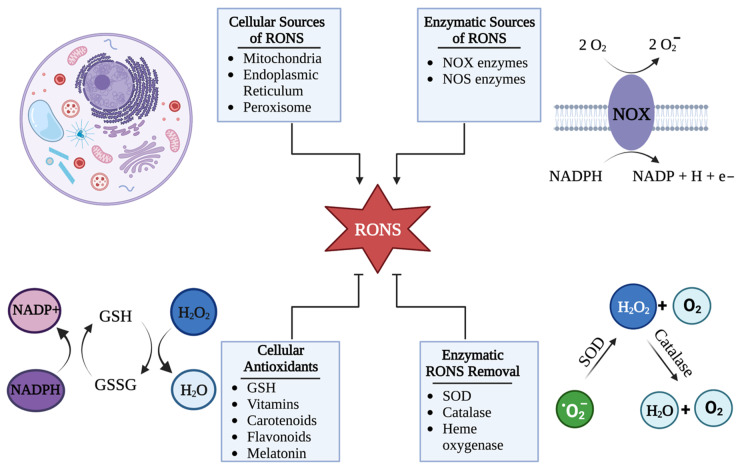
In a cell, RONS can be produced by various organelles during normal metabolic processes including cellular respiration in the mitochondria, protein folding in the ER, and in the peroxisome. Alternatively, enzymes such as NOX and NOS can generate RONS by transporting free electrons across membranes to reduce molecular oxygen. To avoid the accumulation of RONS, cells can oxidize RONS through antioxidants or enzymatically remove them by converting RONS into non-reactive products.

**Figure 4 ijms-24-04673-f004:**
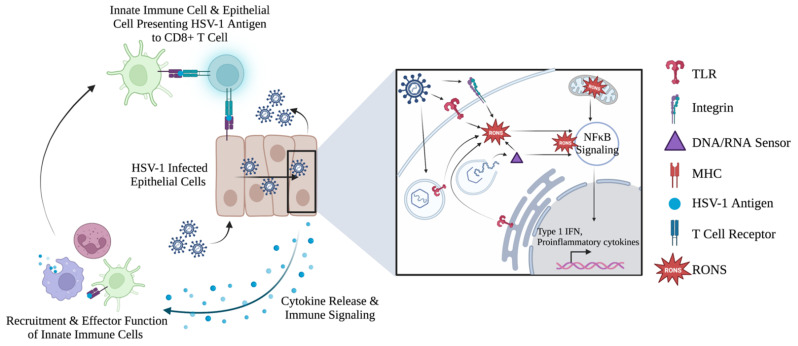
Overview of how cellular immune responses are stimulated during HSV-1 lytic infection. Inside the infected cell, PRRs involving TLRs, DNA/RNA sensors, and integrins sense components of HSV-1. Upon activation of sensors by HSV-1 or HSV-1-induced oxidative stress, immune signaling pathways such as NF-κB can be stimulated to promote the expression of proinflammatory cytokines. When released by the infected cell, these cytokines aid in innate immune cell (e.g., macrophage, dendritic cell, neutrophil) recruitment to promote phagocytosis and uptake of HSV-1 antigens for presentation to the immune system and stimulation of an adaptive anti-HSV-1 immune response (e.g., CD8+ T cells). Alternatively, epithelial cells can also present CD8+ T cells with HSV-1 antigens to prompt elimination of the infected cell.

**Figure 5 ijms-24-04673-f005:**
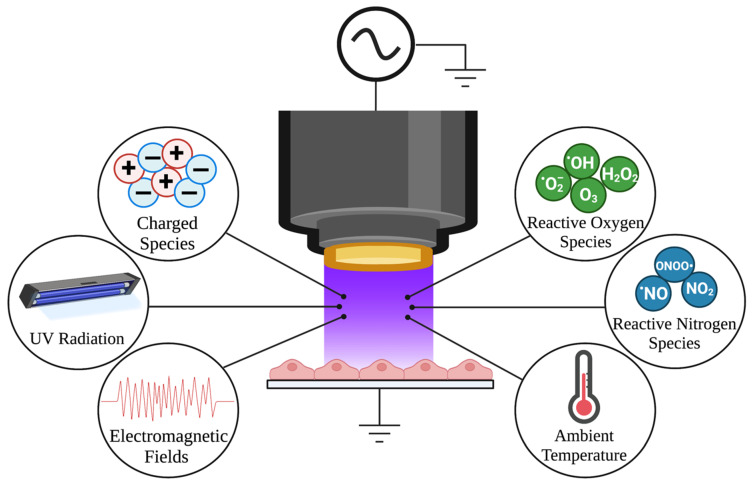
NTP (DBD electrode shown) is a partially ionized gas that generates charged species (electrons, and positive- and negative-charged ions), radiation, electromagnetic fields, and reactive species generated at ambient temperatures. Through collisions between electrons from NTP, and atoms and molecules in the gas and liquid phase, RONS are generated. Following their generation, RONS such as superoxide, nitric oxide, singlet oxygen, ozone, and hydrogen peroxide are delivered to biological targets, which are said to mediate much of NTP’s antiviral effect. The exact composition of RONS delivered can depend on the plasma device and application modality.

**Figure 6 ijms-24-04673-f006:**
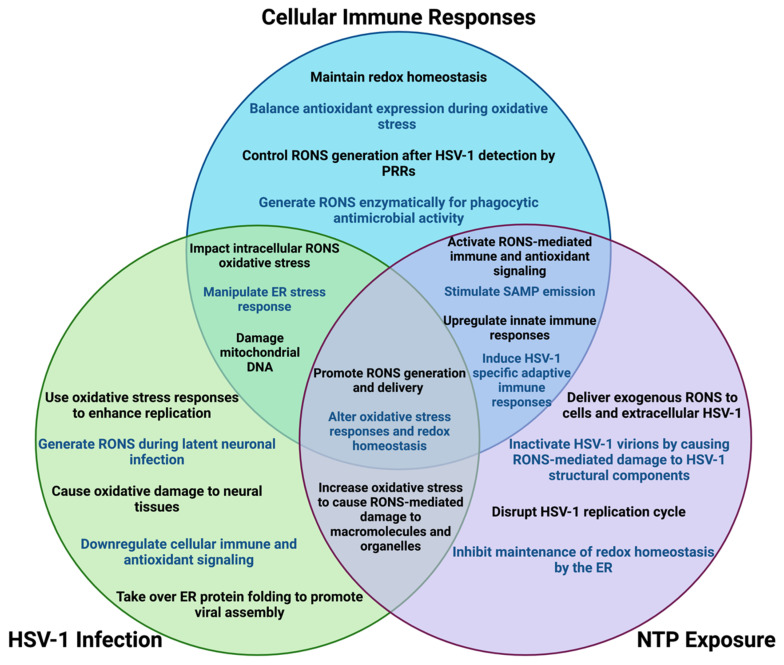
Summary of the roles and effects of oxidative stress in cellular immune responses, HSV-1 infection of host cells, and NTP exposure. The Venn diagram illustrates the unique and overlapping roles of RONS and oxidative stress in each aspect of a putative NTP-based treatment for HSV-1 infection.

**Figure 7 ijms-24-04673-f007:**
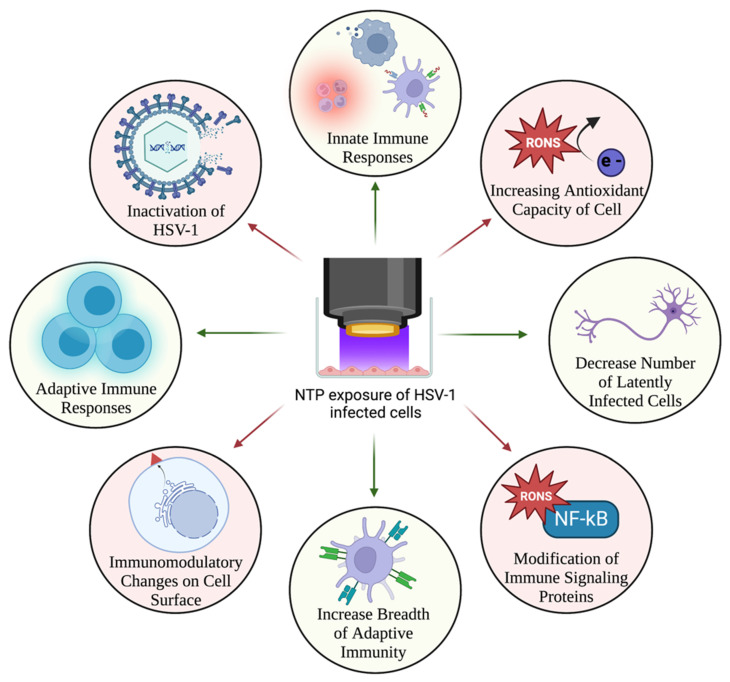
NTP is a potential antiviral and immunomodulatory agent for HSV-1 infection. We propose that the exposure of HSV-1-infected cells to NTP will directly (red) result in the inactivation of HSV-1 virion components and promote the display of SAMPs on the cell surface. Additionally, NTP-generated RONS can directly modify signaling proteins to activate immune signaling and balance redox levels through antioxidant transcription. Based on these effects, we believe NTP will indirectly (green) promote and expand the breadth of stimulation of both innate and adaptive immune responses against HSV-1. In return, this will decrease the amount of virus that can travel into the nervous system to establish viral latency in peripheral ganglionic neurons, ultimately reducing the frequency of reactivation over time.

## Data Availability

Not applicable.

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
