# Peer review of "Manipulation of Oxidative Stress Responses by Non-Thermal Plasma to Treat Herpes Simplex Virus Type 1 Infection and Disease"

_ijms, 2023, doi:10.3390/ijms24054673_

Round 1

Reviewer 1 Report

Excellent Manuscript. I would highly appreciate it if the authors kindly summarize all the information provided in sections 7-10 in a table. It would be easy to understand for the readers. Looking forward to receiving your revised manuscript. 

Reviewer 2 Report

This is a review about the effect of herpes virus and RONS. The subject is interesting, since fight against infection proceeds through formation of RONS by neutrophils etc. and oxidation of viruses. However obviously the authors have not much knowledge in chemistry and understanding of the effects of RONS does require some comprehension of chemical processes and specially of free radicals. Thus some descriptions are simplistic and wrong. They must be corrected before publication. Also, the description of the effect of plasma is totally imprecise. Since this is the heart of this review it must be more informative. In addition, many descriptions are too vague, which should not be in a review.

A list of abbreviations is missing. 

Specific. 

Line 81: RONS never accumulate because their decay is fast either through their reactivity or through enzymatic pathways. Their concentration is always low. Modify the expression.

Lines 101-110: the description of RONS is wrong. Their properties are not attributed by phantasy. they result from their structures. 

Ex.: Line 104-105: Some of them are free radicals (e.g. NO, OH etc.) and some are not (e.g. H2O2). 

Line 105: “derived from oxygen or nitrogen” this is wrong. NOx species are made of nitrogen and oxygen. Except dioxygen, no species has more than one unpaired electron. Thus there is either 0 or 1 unpaired electron.

Line 109: RONS are not “classified” as reactive species, they are reactive species. Again it is not a dream or a will  of chemists, it is a fact.

Line 226: ICP0 acts as ubiquitin ligase, not like ubiquitin (see Rodriguez et al. Virus res. 2020). This is wrong. Modify the sentence.

Lines 293-4: not precise enough: where does the onset of oxidative stress come from? interference with signaling pathways is not informative enough.

Line 375: does not “catalyze superoxide” but its disappearance giving H2O2…

Line 380: mechanisms of action of HO-1 vs. NO does not seem so clear according to the literature. Crosstalk of CO/NO in neurovascular systems is known, not removal of NO by itself. Modify the sentence.

Lines 383-5: GSH has more roles vs. RONS than described here. For instance it is a NO transporter and it scavenges several ROS.

Scheme 3: what is O? it cannot remain that way. Replace by O2 or explain.

Lines 448-450: what is the reaction of RONS with Tyr residues here? The only one I know is oxidation and usually they prevent and not activate phosphorylation because oxidation takes place at the same position as phosphorylation. Add references, precise the mechanism related  to this sentence.

Line 460 RONS, at least the free radical ones do not accumulate. Replace it by ‘increase’ for instance

Line 477: not hydroxy, hydroxyL (add l) radicals are certainly involved since they are the most reactive RONS. Conversely the presence of singlet oxygen is doubtful. Again add references for its presence.

Line 495: “garnered” or gained?

Paragraph 6: the role of ozone as powerful disinfectant agent is well known. That of H2O2 as well. The authors should add hypochlorous acid, as known as the preceding ones. This paragraph is OK for a review but not very informative.

paragraph 7.1-7.2 is not precise enough.

Lines 584-5: give examples of free radicals that are not created naturally and that may be useful in therapies.

Figure 5: what are “neutral species” and “charged species”? again give examples.

 The description is totally vague. Figure 5 is too general. A table with species produced in which conditions and with which electrode would be more precise and more useful.

Line 721/ methionine does not form disulfides! Modify text

Line 723: is MPO a RONS???

Lines 724 726/ many RONS oxidize protein residues. It is known that proteins are major targets of RONS, like nucleic acids, unsaturated lipids….

Line 746: not many RONS penetrate through the membrane. Practically only H2O2 does. As a whole paragraph 9.1 should be rewritten with accurate informations.
